# A Comprehensive Evolutionary Study of Chloroplast RNA Editing in Gymnosperms: A Novel Type of G-to-A RNA Editing Is Common in Gymnosperms

**DOI:** 10.3390/ijms231810844

**Published:** 2022-09-16

**Authors:** Kai-Yuan Huang, Sheng-Long Kan, Ting-Ting Shen, Pin Gong, Yuan-Yuan Feng, Hong Du, Yun-Peng Zhao, Tao Wan, Xiao-Quan Wang, Jin-Hua Ran

**Affiliations:** 1State Key Laboratory of Systematic and Evolutionary Botany, Institute of Botany, Chinese Academy of Sciences, Beijing 100093, China; 2University of Chinese Academy of Sciences, Beijing 100049, China; 3School of Earth Sciences, East China University of Technology, Nanchang 330013, China; 4Laboratory of Systematic & Evolutionary Botany and Biodiversity, College of Life Sciences, Zhejiang University, Hangzhou 310058, China; 5Wuhan Botanical Garden, Chinese Academy of Sciences, Wuhan 430074, China

**Keywords:** gymnosperms, chloroplast, RNA editing, G-to-A, GC content

## Abstract

Although more than 9100 plant plastomes have been sequenced, RNA editing sites of the whole plastome have been experimentally verified in only approximately 21 species, which seriously hampers the comprehensive evolutionary study of chloroplast RNA editing. We investigated the evolutionary pattern of chloroplast RNA editing sites in 19 species from all 13 families of gymnosperms based on a combination of genomic and transcriptomic data. We found that the chloroplast C-to-U RNA editing sites of gymnosperms shared many common characteristics with those of other land plants, but also exhibited many unique characteristics. In contrast to that noted in angiosperms, the density of RNA editing sites in *ndh* genes was not the highest in the sampled gymnosperms, and both loss and gain events at editing sites occurred frequently during the evolution of gymnosperms. In addition, GC content and plastomic size were positively correlated with the number of chloroplast RNA editing sites in gymnosperms, suggesting that the increase in GC content could provide more materials for RNA editing and facilitate the evolution of RNA editing in land plants or vice versa. Interestingly, novel G-to-A RNA editing events were commonly found in all sampled gymnosperm species, and G-to-A RNA editing exhibits many different characteristics from C-to-U RNA editing in gymnosperms. This study revealed a comprehensive evolutionary scenario for chloroplast RNA editing sites in gymnosperms, and reported that a novel type of G-to-A RNA editing is prevalent in gymnosperms.

## 1. Introduction

RNA editing is one of the posttranscriptional processes that changes specific nucleotides at the RNA level [1], which affects most mitochondria and chloroplasts and is essential for the expression of functional proteins [2], the generation of translational initiation or termination codons [3,4], and the stabilization of the secondary structure of introns and tRNAs [5]. Except for marchantioid liverworts, RNA editing has been reported to exist in all land plants [6,7]. In plant organelles, there are two types of RNA editing. Specifically, C-to-U (cytidine-to-uridine) conversion is found in almost all land plants, whereas U-to-C (uridine-to-cytidine) conversion is only observed in hornworts, ferns, and some lycophytes [8,9,10,11,12,13]. To date, more than 9100 plant chloroplast genomes (plastomes) have been sequenced (https://www.ncbi.nlm.nih.gov/genome/browse#!/organelles/ (accessed on 10 August 2022), and RNA editing sites have been reported in at least 1600 species (data from Plant Editosome Database [14] in BIG Data Center [15]). However, among these species, the reported RNA editing sites of more than 1400 species are only used to hypothesize the initiation codons in some genes. Even with the latest published data, RNA editing sites of the whole plastome were experimentally verified in only 21 species (including 1 moss, 3 lycophytes, 5 ferns, 1 gymnosperm, and 11 angiosperms) (Appendix A), seriously hampering the comprehensive study of chloroplast RNA editing.

The number of chloroplast C-to-U RNA editing sites varies greatly in different plants; however, the monophyletic origin of RNA editing has been proposed [16]. Many original editing sites have been subsequently lost due to DNA level mutations, whereas it is rare to gain specific editing sites during land plant evolution [16,17,18,19]. For example, there are more than 3400 events of C-to-U editing in the chloroplast transcriptome of the lycophyte *Selaginella uncinata*, whereas fewer than 200 editing sites were found in angiosperm chloroplasts [18,19,20]. In addition, the number of chloroplast RNA editing sites gradually decreased from basal land plants to angiosperms [17], and chloroplast RNA editing is much more abundant in early branching compared with widely investigated model flowering plants, such as *Arabidopsis*, *Nicotiana*, or *Oryza* [18,19]. Recently, Zhang et al. analyzed chloroplast RNA editing events across 21 plants (6 ferns, 4 gymnosperms, and 11 angiosperms) and confirmed many evolutionary features of chloroplast RNA editing [21]. Plant chloroplast RNA editing sites exist not only in the coding regions but also in tRNAs, rRNAs, introns, and intergenic regions [19,22,23,24,25]. However, previous studies on chloroplast RNA editing in plants mainly focused on protein-coding genes [14]. In addition, the study of Zhang et al. was based on publicly available RNA-seq data [21]. In their study, plastomic sequence and RNA-seq data of the same species were derived from different individuals, with a small number of species sampled from the same lineage. For example, only four species of three families were selected from gymnosperms. Therefore, it is essential to comprehensively investigate the evolutionary trajectory of chloroplast RNA editing in some important land plant lineages.

As the sister group of angiosperms, gymnosperms represent five of the six lineages of seed plants, i.e., Cycadales, Ginkgoales, Gnetales, Pinaceae, and Conifer II (non-Pinaceae conifers or Cupressophyta), which diverged before the Jurassic [26,27]. Previous studies have reported significant differences in the number of RNA editing sites among different gymnosperms [17,28,29,30]. In addition, Saina et al. predicted the number of RNA editing sites in six species from four genera of Podocarpaceae [31], and Wu et al. identified C-to-U RNA-editing sites in *Keteleeria davidiana* and five species from five genera of Conifer II [32]. Recently, the study of Wu and Chaw on mitochondrial RNA editing in gymnosperms reported the information of chloroplast RNA editing sites in one species of Gnetaceae, one species of Pinaceae, and two species of Zamiaceae [33]. However, almost all of the above studies exclusively focused on coding regions. In addition, the relationship between the overall evolutionary pattern of chloroplast RNA editing sites and factors affecting the number of RNA editing sites remains controversial. Therefore, it is of great significance to study the evolution of chloroplast RNA editing in gymnosperms.

In this study, the evolutionary pattern of chloroplast RNA editing sites was investigated in 19 species from all 13 families of gymnosperms based on a combination of genomic and transcriptomic data, and a novel type of RNA editing sites, G-to-A was found to be widespread in gymnosperms. In addition, the characteristics of chloroplast C-to-U RNA editing sites in gymnosperms were compared with those in other land plants. Finally, the correlations between the abundance of RNA editing sites and some important factors, such as plastomic size, GC content, and nucleotide substitution rate, were assessed. This study will shed new light on the evolution of chloroplast RNA editing in gymnosperms, even in land plants.

## 2. Results

### 2.1. Characteristics of Chloroplast RNA Editing in Gymnosperms

The plastomic size, GC content, and gene number (protein-coding genes, rRNA, and tRNA) of 19 gymnosperms are shown in Appendix A. The sizes and GC contents of the gymnosperm plastomes varied greatly, ranging from 109,596 bp (*Ephedra przewalskii*) to 164,957 bp (*Zamia furfuracea*) and 34.7% to 39.7%, respectively. In addition, the number of chloroplast genes ranged from 111 (*Pinus armandii*) to 134 (*Zamia furfuracea*), including 65–88 protein coding genes, 4–8 rRNA genes, and 28–41 tRNA genes.

We identified 15 additional RNA editing sites by RT-PCR. Interestingly, all these additional editing sites belong to G-to-A RNA editing, including 13 in *rrn23*, one in the intergenic region, and one in the CDS region of the *ycf1* gene (Figure 1). In addition, there are some C-to-U editing sites 150 bp upstream or downstream of the same strand, which also confirmed the reliability of the G-to-A editing sites (Appendix A). Some editing sites we identified in *rrn23* seems to be polymorphic in the RNA sequences of some species (Figure 1), which could imply a recent gene transfer event of these chloroplast fragments to the nucleus or mitochondria occurred. However, this kind of polymorphism is similar in different species, i.e., *Ginkgo biloba* and *Taxus cuspidata*, which have diverged more than 300 mya [27]. In addition, we found no polymorphisms at these sites in the DNA-seq reads (Figure 1). Therefore, we still treated these sites as G-to-A RNA editing sites in this study. In total, 1435 chloroplast RNA editing sites were identified from 19 gymnosperms, including 1364 C-to-U and 71 G-to-A RNA editing sites (Table 1).

Based on the recently published phylogeny of gymnosperms [27], we found that the number of chloroplast RNA editing sites varied greatly among different gymnosperm lineages (Figure 2a,b). A large number of RNA editing sites (307, 158 and 275) were identified in *Ginkgo biloba*, *Cycas revoluta* and *Zamia furfuracea*, whereas only 3–7 editing sites were found in Gnetales. In addition, the number of RNA editing sites was roughly the same in different species in each lineage with the exception of Pinaceae (165 and 54 in *Picea* and *Pinus*, respectively) (Table 1). Generally, both types of RNA editing sites were unevenly distributed in gymnosperm plastomes. However, there were significant differences in the distribution patterns of C-to-U and G-to-A editing sites in gymnosperms. First, most C-to-U editing sites were highly efficiently edited, whereas the editing efficiency of G-to-A editing sites was generally low (Figure 3). Second, most of the C-to-U editing sites (1082, 79.33%) were located in the coding region, whereas only 32.4% of the G-to-A editing sites were distributed in the coding region (Table 1, Figure 3). Third, most C-to-U RNA editing events (at least 80%) occurred at the first and second codon positions of the CDS region, resulting in nonsynonymous changes. However, no G-to-A RNA editing sites were found in the coding regions of *Ginkgo biloba*, and more editing sites occurred in the third codon position in Pinaceae and Gnetales. In addition, fewer editing sites were noted in the second codon position compared with those at the first and third codon positions in Conifer II (Figure 3). Finally, G-to-A editing events always caused synonymous changes in gymnosperms except Conifer II and Cycadales (Figure 4a). Fourth, in the nonsilent C-to-U editing sites, at least 50% of editing events converted amino acids from hydrophilic to hydrophobic, and the conversion of serine to leucine was the most common amino acid change. In contrast, all G-to-A editing events did not change the hydrophybicity of amino acids (Figure 4b and Appendix A). Finally, greater than 60% of the −1 position of the editing site was in the C-to-U editing events, whereas A was dominant in the G-to-A editing events (Figure 5).

### 2.2. Variability of Chloroplast RNA Editing Sites in Gymnosperms

Intersection analysis of C-to-U RNA editing sites in the CDS showed that most RNA editing sites were lineage and species specific (Figure 6 and Appendix A). A total of 140, 125, 99, 40, and 3 sites were specific to Cycadales, Ginkgoaceae, Pinaceae, Conifer II, and Gnetales, respectively, whereas only 1 site (*rpl20*-296) was shared in all gymnosperm lineages. In addition, Cycadales and Ginkgoaceae shared 37 sites, whereas Conifer II and Cycadales only had 6 common editing sites (Figure 6). Moreover, no common editing sites were found in Conifer II and Gnetales, whereas there were 30 and 6 shared sites in Cycadales and Pinaceae, respectively (Appendix A).

The ancestral state reconstruction of 1082 C-to-U RNA editing sites in the coding region showed that a large number of gain and loss events of RNA editing sites occurred during the evolution of gymnosperms, including 854 gains and 838 losses (Figure 7). The common ancestor of Ginkgoaceae and Cycadales was predicted to gain 68 RNA editing sites, whereas 27 RNA editing sites were lost in the common ancestor of Pinaceae and Gnetales. In addition, the common ancestors of Conifer II and Gnetales lost multiple RNA editing sites, respectively. In contrast, the common ancestors of Pinaceae and Ginkgoaceae gained 40 and 127 editing sites, respectively, and both gain and loss events occurred frequently in the ancestor of Cycadales.

To explore the variation in RNA editing sites among genes in gymnosperms, we compared the editing site density of genes containing editing sites in at least ten species among all species (Appendix A). The editing site density was expressed as the number of editing sites per 1000 bp. The results revealed significant differences in gene editing levels among different lineages of gymnosperms. In Cycadales and Ginkgoaceae, RNA editing sites were found in all genes, except *psbZ* of *Cycas* and *ndhA* of *Ginkgo*. There were no RNA editing sites in *ndhA* and *ndhE* of Pinaceae and Gnetales because all *ndh* genes were absent in both lineages. In addition, in Pinaceae, RNA editing sites were present in all genes of *Picea* and *Cedrus*, but no RNA editing sites were located in a few genes of *Pinus* and *Abies*. Only *petD* and *rps8* had editing sites across all species in Conifer II, and RNA editing sites were located in different genes in the three species of Gnetales (Appendix A). In addition, *petD* and *rps8* were heavily edited in almost all species in gymnosperms, excluding Gnetales

Based on the analysis of 14 genes containing editing sites in at least ten species in 19 gymnosperm species, we found that more genes experienced RNA editing site gain events in the ancestor of Ginkgoaceae + Cycadales, the ancestor of Ginkgoaceae and the ancestor of Pinaceae, whereas more genes had RNA editing site loss events in the ancestor of Gnetales and the ancestor of Pinaceae + Gnetales. In the ancestor of Conifer II and the ancestor of Cycadales, the number of genes gained and lost RNA editing sites was almost equal. In addition, among these genes, the number and times of loss and gain of RNA editing sites of *rpoB* were much higher than those of other genes, followed by *rpoC2* and *rps8* (Figure 7 and Appendix A).

### 2.3. Factors Correlated with the Number of Chloroplast RNA Editing Sites in Gymnosperms

The Pearson correlation coefficient showed that the number of RNA editing sites was significantly positively correlated with plastomic size (*r* = 0.68, *p* = 0.0013), GC and C content (GC content, *r* = 0.71, *p* = 0.00065; C content, *r* = 0.74, *p* = 0.00032), but insignificantly negatively correlated with *d_N_* (nonsynonymous substitution rates), *d_S_* (synonymous substitution rates), *R_N_* (absolute nonsynonymous substitution rates per branch), and *R_S_* (absolute synonymous substitution rates per branch) (Figure 8).

## 3. Discussion

### 3.1. G-to-A, a Novel Type of RNA Editing Site, Is Common in Gymnosperm Plastomes

In this study, we found that G-to-A, a novel type of RNA editing site, is common in our sampled gymnosperm species. Although the number of G-to-A editing sites is small (only 71) and the editing efficiency is low, the reliability of the G-to-A RNA editing sites was confirmed (Figure 1 and Appendix A). So far, G-to-A RNA editing events were only reported in a few angiosperms, such as *Betula platyphylla* [36] and *Vigna* species [37]. Based on studies of *Betula platyphylla* and *Vigna* species from Leguminosae [36,37], the conversion of G to A occurs in the A-rich region, although most of them are synonymous conversions, which could facilitate three-dimensional folding or interaction of protein products [38,39,40,41,42]. In addition, the G-to-A editing events were likely specific to the plastomes of some specific genera, such as *Vigna*, supporting the idea that RNA editing is a mechanism to correct mutations in the genomic coding sequences [16,43]. In a study by Zhang et al. [21], a small number of G-to-A RNA editing sites were reported in almost all species; however, it is impossible to determine whether these sites were real RNA editing sites or DNA-level mutations between different individuals. In fact, many other types of RNA editing sites were identified in the study by Zhang et al. [21], such as T-to-A and T-to-G, which were not reported in other studies. However, in their final discussion, only C-to-U and U-to-C editing types were kept.

Similar to *Betula platyphylla* and *Vigna* species, the number of chloroplast G-to-A RNA editing events in gymnosperms is very low, and the editing efficiency of most edited sites is relatively low. However, G-to-A RNA editing was present not only in protein-coding genes but also in rRNA genes and intergenic regions and rarely in tRNA genes, which was different from that noted in angiosperms (Table 1, Figure 3). The G-to-A RNA editing sites were found in the coding regions of all sampled species, except for six species, i.e., *Ginkgo biloba*, *Cedrus deodara*, *Gnetum montanum*, *Araucaria cunninghamii, Cunninghamia lanceolata*, and *Taiwania cryptomerioides*, but only in the tRNA genes of *Taiwania cryptomerioides*. Interestingly, in 16 of the 19 gymnosperm species, one or two G-to-A RNA editing sites were identified in the same position of a ribosomal RNA gene located in inverted repeats (IRs), *rrn23* (Appendix A), implying that the origin of the G-to-A conversion potentially occurred before the divergence of gymnosperms.

In addition, the distribution pattern of G-to-A editing sites in protein-coding genes varied greatly in different gymnosperm lineages. All G-to-A RNA editing sites in Cycadales occurred at the first and second codon positions, and no sites were found in *Ginkgo biloba*. More sites in Pinaceae and Gnetales appeared in the third codon position, whereas the G-to-A RNA editing sites in Conifer II mainly occurred in the first and third codon positions (Figure 3). In the non-silent editing sites, G-to-A RNA editing did not change the hydrophobicity or hydrophilicity of amino acids (Figure 4), which made their exchange synonymous in physicochemical terms and seemed to support the importance of G-to-A RNA editing for mutation corrections. Previous studies have shown that an editing activity that does not ‘improve’ the encoded amino acid would not be stabilized by natural selection [44]. Thus, G-to-A RNA editing seems to be dispensable and may be lost in the future. However, considering that G-to-A RNA editing is commonly distributed in gymnosperm plastomes, this type of RNA editing site could be found in more species with chloroplast RNA editing being studied in an increasing number of land plants in the future. In addition, we cannot rule out its function in facilitating three-dimensional folding or the interaction of protein products [38,39,40,41,42]. Thus far, it remains a mystery why G-to-A RNA editing appears. Niavarani et al. confirmed the existence of G-to-A RNA editing sites in humans and reported the first enzyme APOBEC3A (A3A) involved in G-to-A RNA editing [45]. However, in cattle, the homologous genes of A3A showed little or no expression in the mammary gland, although some G-to-A RNA editing sites were also found in this organ [46]. Multiple proteins are required for C-to-U RNA editing in plants [47]. Pentatricopeptide repeat (PPR) proteins represent key characterized RNA-editing factors and play an important role in the recognition of C-to-U RNA editing sites and the subsequent process of cytidine deamination [48,49]. Their PPR domains have a well-defined function in selecting editing sites [50], and the DYW domain has cytidine deaminase activity, which has been recently biochemically confirmed [49,51,52]. However, the direct modification of G to A requires amination at the C6 position and concomitant deamination at the C2 position, which is completely different from C-to-U RNA editing. So far, there is no evidence that PPR proteins can be involved in G-to-A RNA editing. In addition, it is worth noting that we should exclude some other G modifications, especially amination at the C6 position of G to create 2,6-diaminopurine, which seems to mimic A based on the base-pairing properties [53]. 2,6-Diaminopurine is a naturally occurring adenine (A) analog that bacteriophages employ in place of A in their genetic alphabet. Theoretically, 2,6-diaminopurine should not be found in plants, but we cannot completely rule out the possibility of the existence of 2,6-diaminopurine in plants. Especially, analyses of the majority of post-transcriptional modifications, including G-to-A RNA editing and amination at the C6 position of G, lack a well-defined sequencing method. Therefore, additional studies and molecular biological validation are needed to better understand the biological adaptability of G-to-A RNA editing in land plants. In addition, future new sequencing techniques could also verify what happens at those sites in *rrn23* at the transcriptional level.

### 3.2. Chloroplast C-to-U RNA Editing Sites of Gymnosperms Share Many Common Characteristics with Other Land Plants but Also Have Some Unique Characteristics

In this study, we found that the chloroplast C-to-U RNA editing sites of gymnosperms have many common characteristics with other land plants, such as liverworts [54], ferns [8,55], and angiosperms [19,56]. For example, the majority of C-to-U RNA editing events occurred in the first and second codons of protein-coding genes, resulting in nonsynonymous conversions (Figure 3 and Figure 4a). The editing events always led to amino acids changing from hydrophilic to hydrophobic or maintaining hydrophobicity, and the two most prominent amino acid alterations were Ser to Leu and Pro to Leu (Figure 4b and Appendix A). In addition, analysis of nucleotides at the −1 position adjacent to the edited sites showed that greater than 65% of nucleotides at the −1 position were pyrimidines (Figure 5a). Finally, only one site (*rpl20*-296) was shared by all gymnosperm lineages and there were more RNA editing sites shared by the close-related taxa (Figurse 6 and S3), similar to angiosperms [19]. These common characteristics of chloroplast C-to-U RNA editing supported the idea of the monophyletic origin of C-to-U RNA editing in land plants [16]. In fact, some previous studies have shown that the residues encoded by edited codons frequently appeared in the helices, subunit interfaces and protein structural cores [57], and the conversion of Ser to Leu could increase the hydrophobicity of residues located in subunit interfaces and promote subunit interaction. The conversion of Pro to Leu could avoid the situation where proline as an imino acid could break the α-helix and affect proper protein folding [58], therefore suggesting that RNA editing could be functionally and structurally relevant. Nevertheless, the purpose of RNA editing in plant organelles was much more important to the repair of defective transcripts at the RNA level [59]. Multiple RNA editing sites were proved to be probably not critical and essential for *Arabidopsis* growth to maturity but have significant selective advantages for individuals in which environmental conditions are less than optimal, which could be the reason why few RNA editing sites were shared in gymnosperms and angiosperms (reviewed by Stern et al. [60]). In addition, in experiments using transplastomic tobacco plants, Bock et al. found that nucleotide substitution at the −1 position of the edited C severely reduced chloroplast RNA editing efficiency [61]. Therefore, the high proportion of pyrimidine nucleotides at the −1 position implied the importance of this position for RNA editing, and the editing machinery is similar in land plants.

However, some unique characteristics of chloroplast C-to-U RNA editing sites were found in gymnosperms. First, we found that genes with the largest number or proportion of C-to-U RNA editing sites are different in different lineages of gymnosperms. Generally, in almost all samples of Cycadales, Ginkgoaceae, and Conifer II, the number of C-to-U RNA editing sites was the highest in the *ndh* genes encoding subunits of the NADH degydrogenase-like (NDH) complex and the *rpo* genes encoding subunits of the DNA-dependent RNA polymerase, which is similar to that noted in angiosperms [19,56,62,63,64]. However, most C-to-U RNA editing sites were found in the *rpo*, *psb*, or *pet* genes in Pinaceae and Gnetales due to the absence of all *ndh* genes [65,66,67,68]. In addition, similar to liverworts [54], the density of C-to-U RNA editing sites in the *ndh* genes was not the highest in sampled gymnosperms (Appendix A).

In addition, we found that both loss and gain events of C-to-U RNA editing sites occurred frequently in gymnosperms (Figure 7), whereas loss events were predominately noted across angiosperms [19]. *Ginkgo* shared 10 and 14 RNA editing sites with *Cycas* and *Zamia*, respectively, and 13 editing sites with the three species (Appendix A), but only one editing site was shared in all lineages (Figure 6), suggesting the evolutionary conservation of RNA editing sites in the basal gymnosperms. However, significant differences in the evolutionary history of loss and gain of editing sites are noted in different lineages. The extremely low number of C-to-U RNA editing sites in Gnetales could be explained by massive losses of editing sites occurring after divergence from its sister-group lineage (Pinaceae), whereas numerous gains occurred in the two basal lineages, Ginkgoaceae and Cycadales. In addition, multiple loss events (30) occurred in the common ancestor of Conifer II, whereas 40 sites could have been gained in the common ancestor of Pinaceae (Figure 7). Finally, in the 19 sampled gymnosperms, significant differences in the editing site densities of different genes were noted, suggesting that the gain and loss history of RNA editing sites of each gene could be diverse during the evolution of gymnosperms (Figure 7). For example, 12 genes lost but only 1 gained RNA editing sites in the ancestor of Gnetales, whereas 10 genes gained but only 1 lost RNA editing sites in the ancestor of Pinaceae, which implied that these genes could have experienced different selective pressures in different gymnosperm lineages.

### 3.3. Several Factors Could Be Related to Variation in Chloroplast RNA Editing Sites in Gymnosperms

Based on the correlation analysis between the number of chloroplast RNA editing sites and some factors, we found that GC content (*r* = 0.71, *p* = 0.00065), especially C content (*r* = 0.74, *p* = 0.00032), was positively correlated with the number of chloroplast RNA editing sites in gymnosperms (Figure 8b,c). Similar findings were reported in other land plants [21,54,69,70,71]. For example, combined with the observation of *Selaginella* species and the results of some species in other studies, Smith [69] found a positive correlation between GC content and the number of RNA editing sites in chloroplasts and assumed that RNA editing could serve as a genomic buffer to complement T-to-C mutations at the DNA level when facing GC-biased mutation pressure. Therefore, the rise of GC content could provide more materials for RNA editing and facilitate the evolution of RNA editing in land plants or vice versa [54]. In addition, we found a positive correlation between plastomic size and the number of RNA editing sites in gymnosperms (*r* = 0.68, *p* = 0.0013) (Figure 8a). Different degrees of shrinkage and loss of IR regions could contribute to the variation in plastomes in different gymnosperm lineages [72]. Combined with Conifer II, it seemed to have the same number of RNA editing sites regardless of plastome size or GC content; further research is needed to test whether this correlation exists widely in land plants. It was interesting that we found no significant relationship between *d_S_*, *d_N_*, *R_N_*, and *R_S_*, and the abundance of chloroplast RNA editing sites (Figure 8d–g). In fact, a correlation between the number of editing sites and mutation rate has been found in chloroplasts [16] and even mitochondria of some other seed plants [73,74]. However, Sloan et al. suggested that the high rate of C-to-T mutation at editing sites caused by gene conversion with reverse-transcribed mRNA (i.e., retroprocessing) [73], rather than mutations directly at the DNA level, could be related to the rapid loss of editing sites. Therefore, the acceleration of the mutation rate could have not functionally changed the evolutionary focus acting on RNA editing sites.

The correlation between the frequency of RNA editing sites and the diversity of PPR proteins has been frequently reported in plants. Particularly, the evolutionary distribution of organelle RNA editing and the DYW domains seemed to correlate well along land plant phylogeny [6,75,76]. However, except for the DYW type, all five types of PLS PPR proteins were significantly positively correlated with the abundance of RNA editing sites in liverworts [54]. In fact, any plant organelle editosome needs a PPR protein targeting the editing site and a DYW domain bringing cytidine deaminase activity [77]. When the protein belongs to the PPR-DYW subfamily, the DYW domain is provided by the PPR specificity factor. Alternatively, when the specificity factor is a PPR-PLS, PPR-E, or PPR-E+ protein, it can recruit other DYW domains for catalyzing the C-to-U conversion [1,77,78]. However, the number of editing sites affected by a single factor may be as high as seven as demonstrated for SLO2 [79] and OGR1 [80] or even eight as shown for MEF13 [81], and this number may possibly be greater for others [77]. In addition, some nuclear-encoded factors may be dual targeted to chloroplasts and mitochondria [77,82], although the impact of dual targets may be relatively small because only a few cases of RNA editing factors with dual targets have been found in angiosperms. Furthermore, a possible correlation between plastid RNA editing sites and PPR proteins only occurs when the ratio of the number of editing sites between mitochondria and plastids is comparable across species. However, in plants, especially in angiosperms, some species or genera that exhibit highly divergent mitochondrial mutation rates have lost a huge number of mitochondrial editing sites, but there is no significant difference in the editing frequency of chloroplast genes, which lack the mutation rate variation observed in the mitochondrial genome [73]. Finally, the PPR genes are not only critical for organelle RNA editing, but are also involved in RNA processing and translation, trimming of 3′ and 5′ ends, cis- and in some cases trans-splicing, stabilization or destabilization of transcripts, etc. Therefore, the correlation between the number of RNA editing sites and any type of DYW subfamily PPR proteins could need further confirmation in land plants. In particular, the number of PPR proteins were extracted from transcriptomes in some studies, the possibility that not all PPR proteins have been extracted cannot be ruled out, which may lead to some biases in the analysis. Therefore, we did not test the relationships between PPR proteins and RNA editing sites in this study due to that only a few genomes of gymnosperms have been sequenced and the number of PPR proteins identified from one species in different studies varied greatly [33,83].

## 4. Conclusions

The evolutionary pattern of chloroplast RNA editing sites was investigated in 19 species from all 13 families of gymnosperms based on a combination of genomic and transcriptomic data. In total, 1435 chloroplast RNA editing sites were identified from 19 gymnosperms, including 1364 C-to-U and 71 G-to-A RNA editing sites. Interestingly, G-to-A RNA editing events are common in gymnosperms but have many different characteristics from C-to-U RNA editing sites. It still remains a mystery why G-to-A RNA editing appears in plants. Multiple proteins are required for C-to-U RNA editing in plants, but there is no evidence that these proteins can be involved in G-to-A RNA editing because direct modification of G to A requires amination at the C6 position and concomitant deamination at the C2 position, which is quite different from the process of C-to-U RNA editing. We speculated that G-to-A RNA editing sites could have been neglected due to that chloroplast RNA editing has only been studied in detail in a few species, leading to the fact that they were currently only found in a small number of taxa outside gymnosperms. This type of RNA editing sites will be found in more species and the molecular mechanisms behind G-to-A RNA editing will be gradually unraveled with chloroplast RNA editing being studied in an increasing number of land plants in the future. In addition, the chloroplast C-to-U RNA editing sites of gymnosperms share many common characteristics with other land plants, supporting the monophyletic origin of RNA editing sites. However, they also have many unique characteristics, such as the distribution patterns and loss and gain histories in protein-coding genes. Finally, GC content and plastomic size were positively correlated with the number of chloroplast RNA editing sites in gymnosperms, but further research is needed to test whether this correlation exists widely in land plants.

## 5. Materials and Methods

### 5.1. Taxon Sampling and Sequencing Data Processing

In total, 19 species from all 13 families of gymnosperms were sampled (Appendix A). The DNA-seq and strand-specific RNA-seq data of 19 samples were retrieved from National Center for Biotechnology Information (NCBI), which were sequenced in our previous studies [27,34,84,85]. For each species, the materials used for genome and transcriptional sequencing were obtained from young leaves or buds of the same individual, avoiding the possibility of identifying polymorphisms between individuals as editing sites. The raw reads were checked with FastQC (https://www.bioinformatics.babraham.ac.uk/projects/fastqc/, accessed on 5 January 2022). Then, the low-quality reads were filtered and the adaptor sequences were trimmed using Trimmomatic (ILLUMINACLIP:TruSeq3-PE-1.fa:2:30:10 LEADING:20 TRAILING:20 MINLEN:36) [86]. Detailed information is presented in Appendix A.

### 5.2. Plastomic Assembly and Annotation

To ensure assembly reliability, the plastomes of 19 species were assembled using the following approach. First, we applied Bowtie2 v 2.2.9 (Center for Bioinformatics and Computational Biology, Institute for Advanced Computer Studies, University of Maryland, College Park, Washington D.C., USA) [87] to build an index file (“bowtie2-build” command) for each species using the plastome of the same species or its relatives and then extracted the mapping reads (“bowtie2” command). Next, the chloroplast-like reads of each species were assembled de novo into contigs and scaffolds using SPAdes v 3.11.1 (Algorithmic Biology Laboratory, St. Petersburg Academic University, Russian Academy of Sciences, Saint Petersburg, Russia) with *k-mer* values set to 21, 33, 55, 77, 99, and 127 (-k 21, 33, 55, 77, 99, 127, --careful) [88]. Finally, all scaffolds and contigs were mapped to the published plastome of the same species or its relatives using MAFFT v.7 (Immunology Frontier Research Center, Osaka University, Suita, Japan) [89]. Based on the mapping results, multiple primer pairs were designed to fill gaps between scaffolds/contigs (Appendix A).

A BLASTN search was performed to annotate protein-coding genes and ribosomal RNAs (rRNAs) using a local database of extracted gene sequences from published gymnosperm plastomes with evalue and identify sets of 1e-5 and 0.85, respectively, and tRNAscan-SE v.2.0.3 (Department of Biomolecular Engineering, University of California Santa Cruz, Santa Cruz, CA, USA) (cut-off value: 40) was used to annotate the transfer RNAs (tRNAs) [90]. The annotation results were checked manually in BioEdit v7.2 (IBIS Biosciences Inc., an Abbott Company, Carlsbad, USA) [91] and Sequin v16.0 (National Library of Medicine, Bethesda, MD, USA) for the positions of start and stop codons in coding regions and then verified by using GeSeq (BLAT search identify: 85) [92] and PGA (–i 1000 –p 40 –q 0.5,2) [93] with published plastomes of the same species or its relatives as references. Based on the above results, the maps of 19 circular plastomes were drawn by OGDRAW v.1.3.1 (Max Planck Institute of Molecular Plant Physiology, Potsdam-Golm, Germany) (Appendix A) [94].

### 5.3. RNA Editing Site Identification and Analysis

RNA editing sites of all plastomes were identified using RES-Scanner [95], which can use matching DNA-seq and RNA-seq data from the same individuals as input files to detect RNA-editing sites effectively. The editing efficiency (i.e., the ratio of RNA reads containing a specific edited nucleotide site) and the minimum number of DNA and RNA reads for determining each RNA editing site were set with default parameters. To reduce false-positives and negatives, we performed the following steps to verify the editing sites generated by RES-Scanner. First, we selected a species (*Gnetum montanum*) with few RNA editing sites to visualize its bam files (Binary Alignment/Map file: a compressed binary format of Sequence Alignment/Map (SAM) file (.sam), which is used for storing the sequence alignment information) of aligned RNA reads from RES-Scanner using Integrative Genomics Viewer (IGV) v. 2.4.10 (Broad Institute of Massachusetts Institute of Technology and Harvard, Cambridge, USA) [96]. Second, all possible mRNA changes observed from the bam files (including C-to-U, G-to-A, T-to-A, A-to-G, etc.) were verified by reverse transcription polymerase chain reaction (RT–PCR). We found that only C-to-U and G-to-A changes calculated by RES-Scanner were real RNA editing sites. However, we also found that a few RNA editing sites (G-to-A) verified by RT–PCR were not included in the result file of RES-Scanner. Through the comparative analysis of the upstream and downstream sequences of these sites, we found that insertion/deletion occurred at or before/after these sites, which could explain why they were not counted by RES-Scanner (Figure 1). Based on the above results, we verified all candidate editing sites which could be ignored by RES-Scanner of the remaining species. Besides that, we examined the upstream and downstream sequences of G-to-A editing sites to confirm the reliability of the G-to-A editing sites. Finally, all editing sites occurring in genes (e.g., editing type, codon change) were checked manually in Geneious v.2019.0.3 (Biomatters, Ltd., Auckland, New Zealand). In addition, all RNA editing sites were checked in IGV and some C-to-U RNA editing sites identified by RES-Scanner with low RNA-seq read coverage were also verified by using RT–PCR (Figure 1). Therefore, some artefacts unrelated to editing, such as mis-mapping of reads, incomplete trimming of adaptors, sequencing errors, DNA polymorphism and misattribution of the read to the wrong strand can be ruled out in our study. All primers and detailed information of each editing site are listed in Appendix A, respectively.

Intersections among edited sites of coding regions in five lineages and 19 species were evaluated using UpSetR v.1.4.0 (Department of Biomedical Informatics, Harvard Medical School, Boston, USA) [97]. The ancestral state reconstruction of RNA editing sites was inferred using GLOOME (Department of Cell Research and Immunology, Tel-Aviv University, Tel Aviv, Israel) [98] with the Mixture model, which allows to study differences in both gain and loss probabilities. *Amborella trichopoda* [19], *Nymphaea* ”Joey Tomocik” and *Liriodendron tulipifera* [56] were selected as outgroups representing the basal angiosperms.

### 5.4. Correlation Analyses between Some Important Factors and the Number of RNA Editing Sites

Pearson correlation coefficient analysis was employed to test whether some factors, such as plastomic size, chloroplast GC and C content, synonymous and nonsynonymous rates (*d_S_* and *d_N_*), and absolute rates of substitutions per branch (*R_S_* and *R_N_*), were related to the abundance of RNA editing sites. The substitution rates were calculated following the study of Kan et al. [85]. The Pearson correlation coefficient was calculated using the “stats.pearsonr()” function implemented in the SciPy library [99] of Python 3.7.4 (https://www.python.org/ (accessed on 12 March 2022). The scatter diagram and regression line between the number of RNA editing sites and each factor were drawn using the “geom_point()” and “geom_smooth()” functions in the ggplot2 package of R 4.0.2 (https://www.r-project.org/, accessed on 12 March 2022).

## Figures and Tables

**Figure 1 ijms-23-10844-f001:**
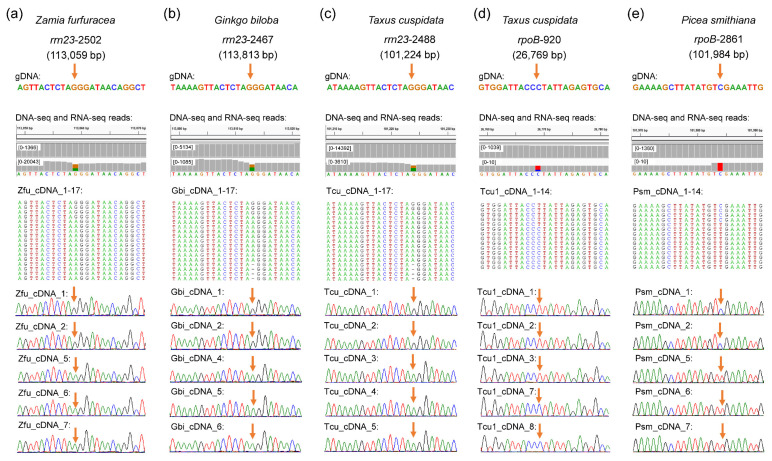
Examples of identification and validation of G-to-A and C-to-U RNA editing sites. The left three panels represent G-to-A RNA editing sites (**a**–**c**), and the last two panels indicate C-to-U RNA editing sites with low RNA-seq read coverage (**d**,**e**). Editing sites are indicated by arrows. From top to bottom shows species name; genic and plastomic positions of RNA editing sites; RNA editing sites and their upstream and downstream sequences; the mapping of DNA-seq and RNA-Seq data on plastomes, numbers on the left-top show read coverage; sequences of cDNA clones; and Sanger sequencing of some examples of cDNA clones.

**Figure 2 ijms-23-10844-f002:**
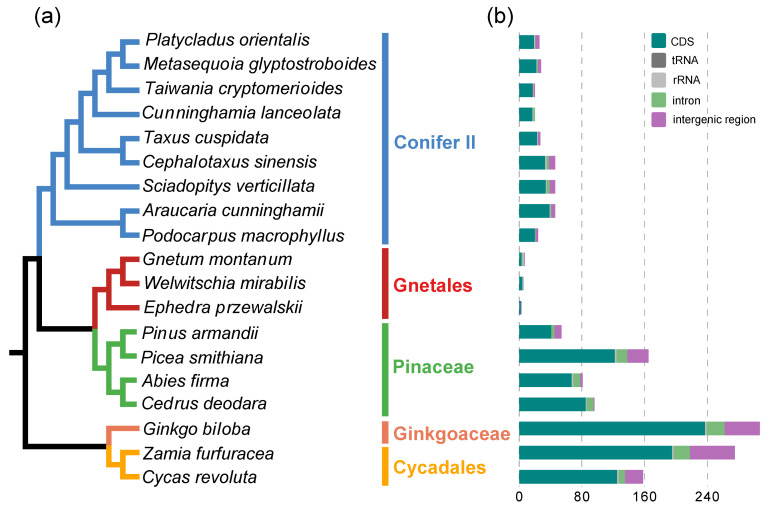
Phylogenetic tree of 19 gymnosperms with associated information of RNA editing sites.
(**a**) Cladogram of 19 gymnosperms based on Ran et al. [27,34]. (**b**) Histogram of chloroplast RNA
editing site abundance across gymnosperms.

**Figure 3 ijms-23-10844-f003:**
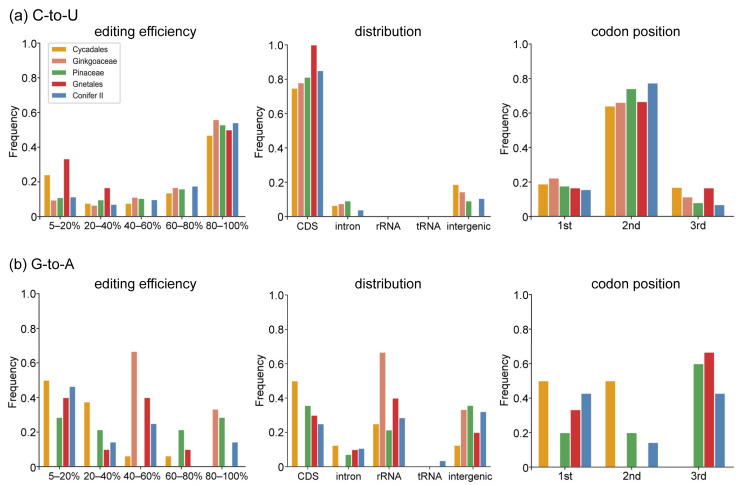
Comparative analyses of editing efficiency, distribution and codon position of chloroplast
C-to-U (**a**) and G-to-A (**b**) RNA editing sites among five gymnosperm lineages.

**Figure 4 ijms-23-10844-f004:**
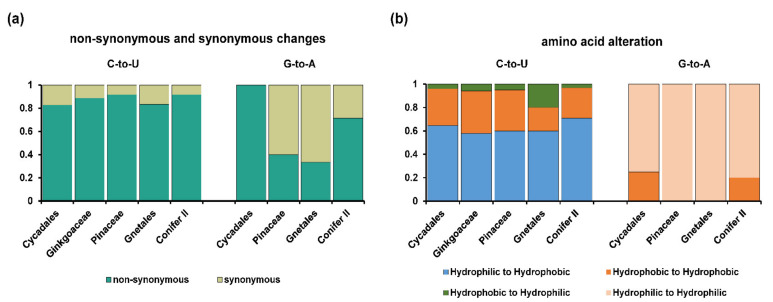
Comparative characterization of RNA editing sites of chloroplast C-to-U and G-to-A RNA
editing sites among five gymnosperm lineages. (**a**) Non-synonymous and synonymous changes; (**b**)
amino acid alteration.

**Figure 5 ijms-23-10844-f005:**
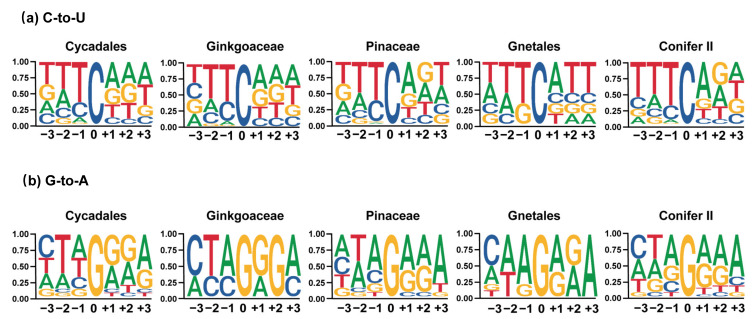
Context (123 sites) flanking the focal C-to-U (**a**) and G-to-A RNA editing sites (**b**) among
five gymnosperm lineages. The sequence logos were generated using the “ggseqlogo” package [35].

**Figure 6 ijms-23-10844-f006:**
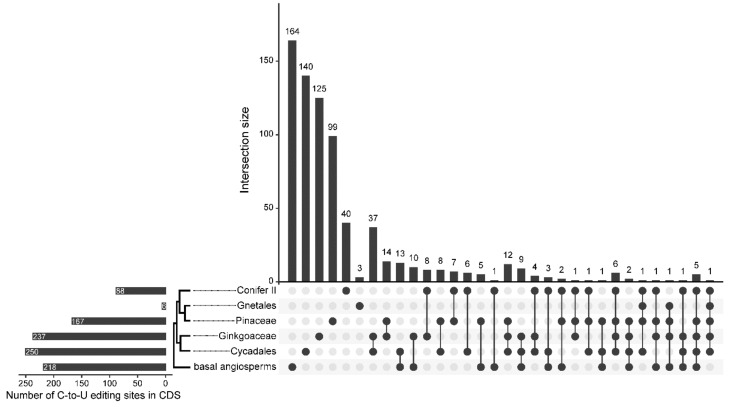
UpsetR plot illustrating the intersection of chloroplast C-to-U RNA editing sites of different
gymnosperm lineages. The bottom-left bar plot shows the number of RNA editing sites in the
CDS.

**Figure 7 ijms-23-10844-f007:**
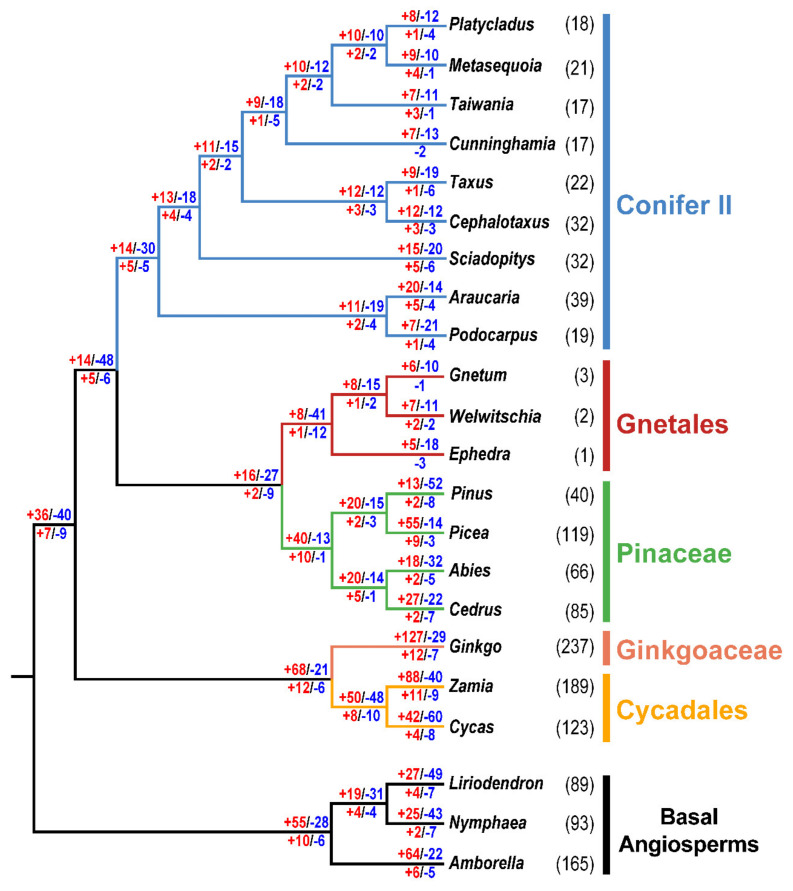
Ancestral state reconstruction of chloroplast C-to-U RNA editing sites during the evolution of gymnosperms. Numbers above the branches denote number of gain and loss events of C-to-U RNA editing sites in the CDS regions, and numbers below the branches are numbers of genes (14 genes containing C-to-U RNA editing sites in at least ten species in 19 gymnosperm species, details see Appendix A) gained and lost C-to-U RNA editing sites, respectively. Values in parentheses are the number of C-to-U RNA editing sites detected in the CDS regions.

**Figure 8 ijms-23-10844-f008:**
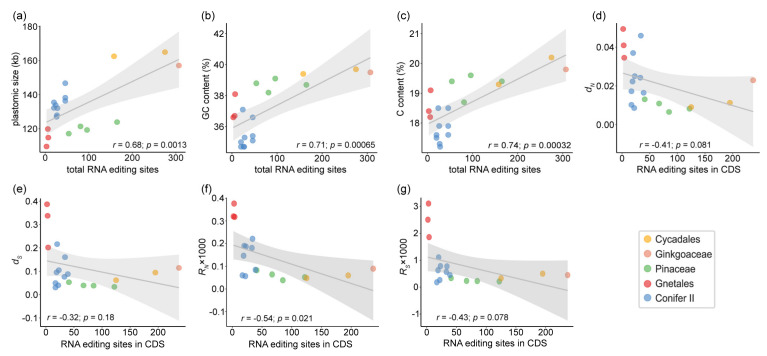
Correlation between the number of chloroplast RNA editing sites and some factors. (**a**) Plastomic size; (**b**) GC content; (**c**) C content; (**d**) *dN* (nonsynonymous substitution rates); (**e**) *dS* (synonymous substitution rates); (**f**) *RN* × 1000 (*RN*: absolute nonsynonymous substitution rates per branch); and (**g**) *RS* × 1000 (*RS*: absolute synonymous substitution rates per branch).

**Table 1 ijms-23-10844-t001:** Summary of chloroplast RNA editing sites in gymnosperms.

Species	Editing Type	Number of Editing Sites	RNA Editing Sites in Coding Regions					RNA Editing Sites in Non-Coding Regions			
			1st Codon	2nd Codon	3rd Codon	Silent Editing	Non-Synonymous Editing	tRNA	rRNA	Intron	Intergenic Region
*Cycas revoluta*	C-to-U	152	23	89	11	13	110	0	0	7	22
	G-to-A	6	1	1	0	0	2	0	2	1	1
*Zamia furfuracea*	C-to-U	265	36	111	42	41	148	0	0	20	56
	G-to-A	10	3	3	0	0	6	0	2	1	1
*Ginkgo biloba*	C-to-U	304	53	157	27	27	210	0	0	23	44
	G-to-A	3	0	0	0	0	0	0	2	0	1
*Cedrus deodara*	C-to-U	95	15	67	3	3	82	0	0	9	1
	G-to-A	1	0	0	0	0	0	0	1	0	0
*Abies firma*	C-to-U	77	10	51	5	5	61	0	1	8	2
	G-to-A	4	0	0	1	1	0	0	1	1	1
*Picea smithiana*	C-to-U	161	23	84	12	12	107	0	1	14	27
	G-to-A	4	1	0	2	2	1	0	1	0	0
*Pinus armandii*	C-to-U	49	7	28	5	6	34	0	0	4	5
	G-to-A	5	0	1	0	0	1	0	0	0	4
*Ephedra przewalskii*	C-to-U	1	0	1	0	0	1	0	0	0	0
	G-to-A	2	0	0	1	1	0	0	0	0	1
*Welwitschia mirabilis*	C-to-U	2	1	0	1	1	1	0	0	0	0
	G-to-A	4	1	0	1	1	1	0	2	0	0
*Gnetum montanum*	C-to-U	3	0	3	0	0	3	0	0	0	0
	G-to-A	4	0	0	0	0	0	0	2	1	1
*Podocarpus macrophyllus*	C-to-U	20	2	17	0	0	19	0	0	0	1
	G-to-A	4	0	0	1	1	0	0	0	1	2
*Araucaria cunninghamii*	C-to-U	45	4	32	3	5	34	0	0	1	5
	G-to-A	1	0	0	0	0	0	0	1	0	0
*Sciadopitys verticillata*	C-to-U	41	7	20	5	5	27	0	0	4	5
	G-to-A	5	2	0	0	0	2	0	1	0	2
*Cephalotaxus sinensis*	C-to-U	40	6	25	1	1	31	0	1	3	4
	G-to-A	6	0	0	1	0	1	0	1	0	4
*Taxus cuspidata*	C-to-U	25	1	19	2	2	20	0	0	0	3
	G-to-A	2	0	1	0	0	1	0	1	0	0
*Cunninghamia lanceolata*	C-to-U	18	3	14	0	0	17	0	0	1	0
	G-to-A	2	0	0	0	0	0	0	1	1	0
*Taiwania cryptomerioides*	C-to-U	18	2	14	1	1	16	0	0	0	1
	G-to-A	2	0	0	0	0	0	1	1	0	0
*Metasequoia glyptostroboides*	C-to-U	24	4	14	3	4	17	0	0	0	3
	G-to-A	4	0	0	1	1	0	0	1	1	1
*Platycladus orientalis*	C-to-U	24	5	13	0	1	17	0	0	1	5
	G-to-A	2	1	0	0	0	1	0	1	0	0
**Total**	C-to-U	1364	202	759	121	127	955	0	3	95	184
	G-to-A	71	9	6	8	7	16	1	21	7	19

## Data Availability

The Accession numbers of chloroplast genomes and the SRA numbers of all DNA-seq and RNA-seq data used in this work were listed in Appendix A.

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
