# Peer review of "A Comprehensive Evolutionary Study of Chloroplast RNA Editing in Gymnosperms: A Novel Type of G-to-A RNA Editing Is Common in Gymnosperms"

_ijms, 2022, doi:10.3390/ijms231810844_

Round 1
Reviewer 1 Report
The manuscript “A comprehensive evolutionary study of chloroplast RNA editing in gymnosperms: A novel type of G-to-A RNA editing is common in gymnosperms” has few new insights towards understanding the evolutionary pattern of chloroplast RNA editing sites. The manuscript is well written, interesting results and suitable discussion. I believe the ms could be considered to be published in IGMS after author provide further evidence in G-to-A editing events and modify some minor places.
This study based on the DNA and RNA sequences of 19 species from all 13 families of gymnosperms to investigate the evolutionary pattern of chloroplast RNA editing sites. Although C-to-U RNA editing sites of gymnosperms shared many common characteristics with those of other land plants, some interesting results were put forward. The density of RNA editing sites in ndh genes was not the highest in the sampled gymnosperms, and both loss and gain events at editing sites occurred frequently during the evolution of gymnosperms. The most important result is G-to-A RNA editing events with many different characteristics from C-to-U RNA editing were found in gymnosperm species. These results can help us to better understanding evolutionary scenario for chloroplast RNA editing sites in gymnosperms. However, there are still some points as below may be addressed further.
1. G-to-A RNA editing sites were rarely found in the organelle genome of land plants. This study confirmed many G-to-A RNA editing events by compared the DNA-seq with strand-specific RNA-seq data of 19 samples using RES-Scanner. The author just selected a species (Gnetum montanum) with few RNA editing sites to visualize its bam files of aligned RNA reads and verified possible mRNA changes by RT-PCR in the species, verified all candidate editing sites which could be ignored by RES-Scanner in the remaining 18 species and identified some C-to-U RNA editing sites with low RNA-seq read coverage by using RT–PCR. In other 18 species, the G-to-A editing events calculated by RES-Scanner still need to verified by RT-PCR, which is important to identify these special editing sites.
2. Fig.6 should be moved forward as Fig.1., and the corresponding description of editing sites identification should appear in the results section instead of place all of them in the Materials and Methods part.
3. The analysis of gain and loss of C-U RNA editing sites is also interesting. It would be great if further investigated the different chloroplast genes to see there is any difference in gain and loss editing sites. And how about the gain and loss of G-A editing sites?
4. Some details should be correct, such as page 2, second paragraph, 12 line, “based on publicly available RNA-seq data” repeat with “In addition, the study of Zhang et al. was based on publicly available RNA-seq data” on the 16 line. Delete one sentence would much better.
5. Page 4, 2.2 part, the second sentence, “A total of 140, 125, 99, 40, and 3 sites were specific to Cycadales, Ginkgoaceae, Pinaceae, Conifer II, and Gnetales, respectively, whereas only one site was shared in all gymnosperm lineages.” The only one shared editing site should be indicated in which gene and occurred in which position. In the discussion part, the reason may be briefly discussed.
6. Page 8., the Note of figure 4., “The number with an asterisk indicates the number of editing sites present in the common ancestor of seed plants.” Actually, no asterisk can be found in the figure 4. Please check the data.
7. Page 14., 12 line, Figure S5 maybe write in the wrong place, 19 circular pastimes is Figure S6. Please check.
Reviewer 2 Report
This manuscript by Kai-Yuan Huang and colleagues is a focused characterization of chloroplast RNA editing in gymnosperms that highlights the unusual abundance of G-to-A editing sites in that lineage. Many past works focused on angiosperms, including only a handful of gymnosperm species as token outgroups. This evolutionary cross-section of chloroplast RNA editing in the major gymnosperm groups (Cycadales, Ginkgoaceae, Pinaceae, Gnetales, and Conifer II) highlights the many similarities, but also the key differences, between gymnosperms and the well-studied angiosperms. The dissection of G-to-A editing sites and how they differ from the more common C-to-U editing sites, such as reduced occurrence in protein coding sequence and tendency to not change hydrophobicity of encoded amino acids at the sites that are in a CDS, is a particularly novel contribution to our understanding of plant organelle editing.
Although this submission is a worthwhile contribution to plant RNA editing, there are a few broad and several detailed issues that must be addressed. The lack of discussion on the trans factors of chloroplast RNA editing and any possible connection to G-to-A editing stands out. The review on plant organelle RNA editosomes by Sun T. et al. (2016) provides a description of the proteins involved in RNA editing, and the review of chloroplast RNA metabolism by Stern, D.B. et al. (2010) contains some insights into the functional consequences of RNA editing that would be worth mentioning in the discussion. A bit of speculation on what the chemical reaction is that creates the G-to-A edit would be good. For C-to-U editing, a deamination reaction performed by a PPR protein effects the edit; what may be happening for G-to-A?
Following are specific comments organized by page number, section, and/or figure/table:
· Pg. 2: In the last paragraph of the Introduction, there should be some allusion to abundance of G-to-A editing in gymnosperms.
· Table S1: Why do you list Welwitschia mirabilis as having no editing sites? Were you the first to find these editing sites, in this paper? Clarify if this table is based on prior studies only. It would be more useful to the filed to include your own findings in this table.
o Several species have an outdated number of editing sites listed. Arabidopsis has 43 chloroplast editing sites (Ruwe, H., Castandet, B., Schmitz-Linneweber, C. & Stern, D.B. 2013. Arabidopsis chloroplast quantitative editotype. FEBS Letters, 587, pp. 1429–1433), and P. patens has two chloroplast RNA editing sites (Miyata, Y., and Sugita, M. 2004. Tissue- and stage-specific RNA editing of rps14 transcripts in moss chloroplasts. J. Plant Physiol., 161, pp. 113–115.). A small side note is that that the generic name Physcomitrella has recently been discarded in favor of Physcomitrium (Rensing, S.A, Goffinet, B., Meyberg, R., Wu, S.Z, & Bezanilla, M. 2020. The Moss Physcomitrium (Physcomitrella) patens: A Model Organism for Non-Seed Plants. The Plant Cell, 32, 5, pp. 1361–1376, https://doi.org/10.1105/tpc.19.00828).
o All references in the table have “[J]” after the title, which looks like an error from an automatically generated citation that should be removed.
· Fig. 2: The font is small and difficult to read. I can see that the authors wanted to be able to represent these different characteristics of RNA editing with a consistent color scheme for each lineage, but the end result is that the figure is difficult to interpret without bouncing back and forth between legend and panel, and ultimately, some of these data would be better represented with a visualization other than a bar graph. For example, a series of SeqLogos for each lineage would be a better way to show Fig 2f (five SeqLogos for C-to-U editing, five for G-to-A editing). SeqLogo is a Bioconductor package for R, but I believe a port of the package exists for use in Python as well. Panels d and e would be more easily interpreted as stacked bar graphs with the lineages on the x-axis, rather than separating the non-synonymous/synonymous and hydrophilic/hydrophobic values into separate bars. The authors may find it easier to split Figure 2 into a few figures to improve readability and to emphasize the differences in these characteristics of editing.
· Table 1: I understand if MDPI’s formatting rules disallow it, but some way of making the table easier to read across, such as adding extra space between the pair of rows for each species or alternating row shading, would be preferred. Also, since the table has to split across pages, a repetition of the header rows on the second page would be preferred.
· Pg. 7, Section 2.2: The wording of, “In Conifer II, RNA editing sites were found in only two genes, petD and rps8, in all species…” confusingly reads as though RNA editing sites were only found in these two genes. Could this be reworded as, “Only petD and rps8 had editing sites across all species in Conifer II…”?
· Figure S3: Why does Ephedra appear to have no C-to-U editing sites? Table 1 shows it has an editing site in the intergenic region, but this is not reflected in Figure S3.
· Figure 4: The tree would be improved by color-coding the branches to match the lineages, as was done in Fig. 1a. The figure legend mentions an asterisk indicating editing sites in the seed plant ancestor, but I don’t see an asterisk in the image. Also, the legend does not tell what the numbers in parentheses indicate; Assuming that this tree’s style is similar to that in fig. 4a of Wu C. and Chaw S. 2022, I suppose that those numbers are the number of editing sites detected in the shared genes?
· Pg. 8, Section 2.3: This is the first mention of the substitution rate statistics ds, dN, RN, and RS, but this looks like the only place in the paper where the definitions of those terms is not spelled out. Please define them here and in Figure 5’s legend, then you do not have to re-define them in the discussion.
· Pg. 9, Section 3.1: Why is the OGDRAW paper [71] cited at, “Although the number of G-to-A editing sites is small [71]…”? Is this a mistake?
o How are you defining rrn23 as polymorphic? The statement is not followed by a figure or reference citation; do you mean polymorphic in the gene sequence? Also, how would this polymorphism imply that a recent gene transfer from plastid to nucleus/mitochondria occurred? Where’s the evidence for that?
o “…the conversion of G to A occurs in the A-rich region, which could facilitate three-dimensional folding or interaction of protein products.” Almost this exact sentence was used in the Nawae W. et al. (2020) paper on Vigna G-to-A editing, but their explanation was also lacking. What is it about RNA G-to-A editing that would affect folding or protein-protein interactions?
· Figure 6: Make 6f a supplemental figure, as it does not fit in with the discussion section 3.1. Aside from 6f, the rest of the panels are small and completely unreadable; they need to be scaled up. I recommend filling the page with them and reorganizing the panel designation to match each editing site breakdown (e.g., panel a can be Zumia furfuracea rrn23-2502, panel b can be Gingko biloba, etc.).
· Pg. 10, Section 3.1: Is the entire of Table 2 necessary to make the point that rrn23 in inverted repeats have low editing efficiencies? Perhaps it would be suitable as a supplemental figure.
o Table 2: The second sentence of the table’s title seems inaccurate and redundant; should be removed. Why is the “before alignment” column necessary? It does not seem to contribute anything to the data.
o Why does low editing efficiency of rrn23 imply that the G-to-A mutation occurred before the gymnosperm divergence? Are you saying that the inversion happened before the divergence, or after? Explain this point.
· Pg. 11-12, Section 3.1: Why does the lack of hydrophobicity changes due to G-to-A editing support the importance of G-to-A editing for mutation corrections? Also, hydrophobicity is not the only important characteristic of an amino acid; sterics can make a structural difference as well. If the editing is really “synonymous in physicochemical terms,” why did the editing site evolve in the first place?
o The citation of the cattle G-to-A RNA editing does not seem relevant if there is no further discussion of the trans factors involved in RNA editing.
o What is meant by “excluding” other G modifications? Excluding from what, your analysis?
o What is meant by “biological adaptability of G-to-A RNA editing”?
· Pg. 12, Section 3.2: The statement, “…the majority of C-to-U RNA editing events occurred in the first and second codons of protein-coding genes, resulting in nonsynonymous conversions” seems like it should have a figure citation.
o In the last paragraph on page 12, you mention that only one site was shared in all lineages. What site was it?
· Pg. 13, Section 3.2: “…significant differences in the editing levels” is incorrect; the density of editing sites is what Figure S3 shows, not editing levels (meaning the percent of editing for a given site).
o Expand upon, “…could be diverse during the evolution of gymnosperms.” Are you saying that the genes have been under different evolutionary pressures in each lineage?
· Pg. 13, Section 3.3: The list of citations for “similar findings are reported in other land plants” has the glaring omission of no Arabidopsis papers.
o The speculation on rising GC content leading to more RNA editing is possible, but the correlation in Fig. 5a seems weak, especially as the Conifer II lineage seems to have about the same number of editing sites regardless of plastome size or GC content. This should be addressed here.
o “retroporcessing” is a typo of “retroprocessing.”
o There should be some conclusion paragraph, hopefully addressing possible explanations for why gymnosperms seemed to have retained more of this G-to-A RNA editing, as well as touching upon potential trans actors responsible for the editing.
· Table S4: “Vouchers” does not seem to be the right word for that column; “Vendor” or “Source” would be a better word choice. Please be more detailed for the source of Welwitschia than, “bought from Internet.” From what website? Also, your citation style for the DNA- and RNA-Seq sources doesn’t match the numbered format used in the paper, so please correct that.
· Figure S4: Please spell out IGV in the figure legend and increase the font size of the base pair counter in the image.
· Figure S5: This figure is incorrectly cited in the OGDRAW methods section (pg. 14; plastomes are currently listed as Figure S6). The content of Figure S5 does not seem to be mentioned in the manuscript; either remove it or cite it in the appropriate location.
· Figure S6: The small size of each plastome diagram makes the text impossible to read and the GC/AT content circles very hard to see. It is also clear that some of these diagrams were “squashed” to fit. Since this is a supplemental figure without space restrictions, you might as well expand the size of each diagram to fill the width of the page rather than forcing them into two columns. Additionally, the color legend should be on the first page of this figure.
· Pg. 14, Section 4.3: Please explain briefly what a “bam file” is or what kind of data it contains (a sequence alignment, correct?).
Reviewer 3 Report
The manuscript presented by Kai-Yuan et al. is of good importance for the plant evolution concept. The manuscript needs major revision and here are my. point to be addressed. In the abstract, I don't see the importance of these sentences. "RNA editing challenges the central dogma of molecular biology by changing genetic information at the transcriptional level; therefore, the evolutionary pattern of RNA editing has attracted increasing attention. Although more than 9,100 plant plastomes have been sequenced, RNA editing sites of the whole plastome have been experimentally verified in only approximately 21 species, which seriously hampers the comprehensive evolutionary study of chloroplast RNA editing." Please go directly to your work even though a small introduction will be needed.
The introduction is well designed please cross-check some typos in the text.
In the result section no reference to Figure 1A. explained this and what characterized each of the four groups. Figure 2 quality is poor and too difficult to read. Provide high-quality figures.
I am having concerns about Table 1 regarding the tRNA, the rRNA, and the Intron number reported in this document. This may be due to the methodology used. Why did the authors leave figures and tables in the discussion part? Also, the discussion section needs improvement as some components are common knowledge and don't need to be highlighted in this section. In the materials and methods section authors need to clarify if biology replication was considered for the sampling if not how and what will be the implication to this study as the expression vary from plant to plant and based on the time this need to be addressed. in the assembly and the annotation please state how each software was used including the default parameter.
Reviewer 4 Report
Recommendations for Authors (will be shown to authors) The following questions do not substitute for specific comments made for authors. Please give further details in the comments for authors box below.Yes | Can be improved | Must be improved | Not applicable | |
Does the introduction provide sufficient background and include all relevant references? | ||||
Are all the cited references relevant to the research? | ||||
Is the research design appropriate? | ||||
Are the methods adequately described? | ||||
Are the results clearly presented? | ||||
Are the conclusions supported by the results? |
* English language and style |
Extensive editing of English language and style required Moderate English changes required English language and style are fine/minor spell check required I don't feel qualified to judge about the English language and style |
Yes | No | |
Do you have any potential conflict of interest with regards to this paper? | ||
Did you detect plagiarism? | ||
Did you detect inappropriate self-citations by authors? | ||
Do you have any other ethical concerns about this study? |
Ratings | High | Average | Low | No Answer |
* Originality / Novelty | ||||
* Significance of Content | ||||
* Quality of Presentation | ||||
* Scientific Soundness | ||||
* Interest to the readers | ||||
* Overall Merit |
Author Response
This reviewer did not provide detailed comments and suggestions, but he/she seemed to think that the introduction, methods, results and discussions needed to be improved to some extent. Therefore, we read this manuscript carefully and modified some details. At the same time, the manuscript has been revised in detail according to the comments and suggestions of other three reviewers.
Round 2
Reviewer 1 Report
The modification of MS is much better than the old version. Few suggestion as below.
1. The table 1 should be put in the frist part of result, after the Figure 1.
2. P4. lines 144-145, the sentence "In total, 1,435 chloroplast RNA editing sites were identified from 19 gymnosperms, 144 including 1,364 C-to-U and 71 G-to-A RNA editing sites (Table 1)" could be moved to the previous paragraph.
3. P5. lines 421-428. The meaning expressed in the former sentence,“......,therefore suggesting that RNA editing could be funtionally and structurally relevant” is contradictory to that expressed in the latter sentence,“Multiple RNA editing sites were proved to be probably not critical and essential for Arabidopsis growth to maturity,.......” It should be keep consistance or further disscusion.
Author Response
Responses to Reviewer 1:
The modification of MS is much better than the old version. Few suggestion as below.
- The table 1 should be put in the frist part of result, after the Figure 1.
>>> Thanks for the good suggestion. In the revision, it has been moved to the first part of result (after Figure 1).
- P4. lines 144-145, the sentence "In total, 1,435 chloroplast RNA editing sites were identified from 19 gymnosperms, 144 including 1,364 C-to-U and 71 G-to-A RNA editing sites (Table 1)" could be moved to the previous paragraph.
>>> Thanks for the suggestion. Following the suggestion, this sentence has been moved to the end of the previous paragraph in the revision.
- P5. lines 421-428. The meaning expressed in the former sentence,“......,therefore suggesting that RNA editing could be funtionally and structurally relevant” is contradictory to that expressed in the latter sentence,“Multiple RNA editing sites were proved to be probably not critical and essential for Arabidopsis growth to maturity,.......” It should be keep consistance or further disscusion.
>>> Thanks for the good suggestion. After a careful reading of the reference, the second sentence has been changed to “Multiple RNA editing sites were proved to be probably not critical and essential for Arabidopsis growth to maturity but have significant selective advantages for individuals in which environmental conditions are less than optimal, which could be the reason why few RNA editing sites were shared in gymnosperms and angiosperms (reviewed by Stern et al. [60]).” in the revision, which is consistent with other discussions.
Reviewer 3 Report
The authors have addressed well my queries and have no more comments.
Author Response
Thanks for the positive comments.